# Fire Effect and Performance of Bridge Pylon Columns under Construction

**Yang Li [1], Zuocai Wang [1,2,*], Changjian Wang [1,3], Yin Zhang [1], Hongsheng Ma [1] and Lili Liu [1]**

[1] College of Civil Engineering, Hefei University of Technology, Hefei 230009, China;
2020010050@mail.hfut.edu.cn (Y.L.); chjwang@hfut.edu.cn (C.W.); 2020020011@mail.hfut.edu.cn (Y.Z.);
mahs@mail.hfut.edu.cn (H.M.); 2023800054@hfut.edu.cn (L.L.)

[2] Anhui Province Infrastructure Safety Inspection and Monitoring Engineering Laboratory, Hefei 230009, China

[3] Anhui International Joint Research Center on Hydrogen Safety, Hefei 230009, China

[*] Correspondence: wangzuocai@hfut.edu.cn

**Abstract:** The fire effect and performance of bridge pylons under construction were investigated via an analysis conducted on two types of pylons with different wall thicknesses. Three fire scenarios, namely internal fire, external ring fire, and external side fire, were established for a 40 m high section of the bridge pylon under construction. The distribution of fire smoke and temperature was obtained using fire dynamics simulation software for different fire scenarios. In addition, a finite element simulation was performed using the thermal–mechanical coupling method to obtain the temperature, stress, and deformation of the columns. The simulation results demonstrate that the average temperature of the internal fire is higher. The chimney effect extends the height range of temperature influence. In the vertical direction, the temperature decrease curve for the internal fire follows a single negative exponential function, while the external fire adheres to a double negative exponential function. The thickness of the temperature influence in the bridge pylon is extended by heating to approximately 200 mm. The stress value considering the thermal expansion coefficient is nearly 27.5 times that without the expansion coefficient, while the deformation value increases by 1 to 8 times. In conclusion, the calculations of the coupled expansion coefficient are helpful in improving the fire safety of bridge pylons.

**Keywords:** bridge pylon; columns; fire; concrete; temperature field; thermal–mechanical coupling; thermal expansion coefficient

## 1. Introduction

As urban infrastructure, the structural safety of bridges cannot be ignored. Among the dangerous factors affecting the bridge structure, the fire factor [1,2] is not easy to be concerned with due to its contingency. In recent years, fires have often been caused by different reasons in bridge engineering. The research on bridge fire accidents has been gradually paid attention to. Alos-Moya et al. [3,4] conducted an open fire test on a 6 m long reinforced concrete bridge model. The temperature and deformation are measured to serve as a reference for the simulation and fire resistance design. Aziz et al. [5] used the fire test as the temperature load to track the fire response of the steel bridge beam in the finite element model. Mahamid [6] considered the fire rating of bridge damage under ASTM E119 fire and hydrocarbon fire. An et al. [7] studied the fire temperature field changes in double-deck bridge spacing and proposed the maximum temperature prediction equation of truss with different deck spacing. Kodur et al. [8–10] assessed the fire risk via finite element analysis and calculation and provided a practical and effective fire prevention strategy scheme for engineering personnel.

Long-span bridges, cable-stayed bridges, and suspension bridges have been paid more and more attention to for structural safety under fires [11,12]. Nariman [13] verified the damage and fatigue of cable-stayed bridges using the fluid–structure coupling

and thermal–mechanical coupling analysis methods. Zhang et al. [14] conducted a comprehensive numerical calculation of the Chishi Bridge fire accident to analyze the static performance of the bridge during the damage, failure, and subsequent repair of the stay cable. Xu and Liu et al. [15,16] proposed a method combining computational fluid dynamics and finite element methods to reproduce a real fire combustion process and predict the fire temperature field and thermodynamic response of a long-span cable-stayed bridge.

Bridge pylon structure is an important part of long-span bridges. Sometimes, fire damage to the bridge pylon structure will endanger the stability of the bridge. Cui et al. [17] studied the behavior of a three-tower suspension bridge tower under the influence of a tanker fire using heat transfer analysis and structural stability analysis methods. Ma et al. [18] analyzed the temperature effect and deformation behavior of steel pylons under different fire accidents. However, the fire situation of the bridge pylon under construction is not involved in these studies. In the process of bridge pylon construction, fire may be caused due to the existence of various potentially dangerous fire sources [19–21], such as wood formwork [22], thermal insulation foam [23], paint coating [24], electrical equipment [25], welding [26], etc. A similar fire incident occurred on a highway bridge under construction in California on 6 May 2014. The cause of the accident was that the wooden scaffold supporting the new bridge ignited in flames from a builder's welding gun. The fire burning speed was extremely fast, and the supporting structure around the bridge was quickly engulfed in fire. The bridge, which cost USD 32 million, collapsed only a few minutes after the fire. The accident was the alarm bell for the fire precautions of the construction bridge.

The pylon column structure of the bridge is generally composed of concrete. Therefore, some research on the fire resistance of concrete columns has important reference significance. Gernay [27] quantified the fire resistance data of reinforced concrete columns and established prediction equations to estimate fire resistance. Han et al. [28] obtained the influence rules of fire duration, cross-section size, slenderness ratio, and concrete strength on the fire residual strength of reinforced concrete columns using numerical simulations. Wang et al. [29] studied the effects of the slenderness ratio, load, concrete strength, cross-section size, and reinforcement ratio on fire behavior and performance of concrete-filled steel tube (CFST) columns. Van et al. [30] proposed three probability methods to prove the reliability and safety of concrete columns under fire.

The research methods of bridge fire are mainly fire tests [3–5] and numerical simulations [11–16]. However, it is difficult to implement the test due to factors such as the difficulty of the test and poor measurement accuracy. The limitation of test conditions can be avoided by numerical simulation. The existing numerical simulation methods of bridge fire response are mainly based on the initial ambient temperature conditions and the principle of heat transfer to analyze the heat transfer law and then obtain the structure temperature field. Subsequently, the response of the structure under fire is determined using thermal–mechanical coupling analysis and simplified performance models.

This study investigates the specific fire conditions of bridge pylons during the construction phase, using thermal insulation foam material as the selected fire source. The analysis focuses on evaluating the fire effects on two different types of bridge pylon columns. Temperature distributions within the bridge pylon are obtained under scenarios involving internal fire, external ring fire, and external side fire. By utilizing average temperature data from stable sections, non-uniform temperature distribution models are established, accounting for temperature variations with height. These models are then used to apply average temperatures at different heights as non-uniform temperature loads on the surface of the pylon column under construction. Consequently, the thermodynamic behavior characteristics of the pylon column are determined. This analytical approach provides valuable insights for guiding effective fire protection measures for pylon structures at bridge construction sites.

## 2. Fire Scenarios of Bridge Pylon during Construction

### 2.1. Fire Scenarios Assumption

During the construction of cable-stayed bridges, pylon fires can be caused by a variety of accidental factors. Especially in the welding operation, the combustibles on the pylon under construction would be ignited by careless flame and sparks. In this paper, a cable-stayed bridge is taken as an example. There are two types of bridge pylons: the side pylon (SP) and the main pylon (MP). In order to reduce self-weight, the pylon columns are hollow, as shown in Figure 1. The wall thickness of the SP and MP pylon are 1.4 m and 1.75 m, respectively. A 40 m high column section of bridge pylons under construction is selected for fire analysis. The length and width of outer surface in SP are 12.6 m and 16 m, respectively, while those of MP section are 12.8 m and 13.6 m, respectively. In terms of steel reinforcement settings, longitudinal bars and stirrups are mainly considered, with a protective layer thickness of 150 mm. A 32 mm HRB400 is applied to longitudinal reinforcement, while $\Phi$20 mm HRB400 is used for stirrups. Three fire scenarios are proposed for the SP and MP, respectively, as shown in Table 1. Scenario I is a fire of the accumulation inside the bridge pylon column. Scenario II is a ring-shaped accumulation fire outside the bridge pylon column. Scenario III is a fire caused by accumulation on one side outside the bridge pylon column.

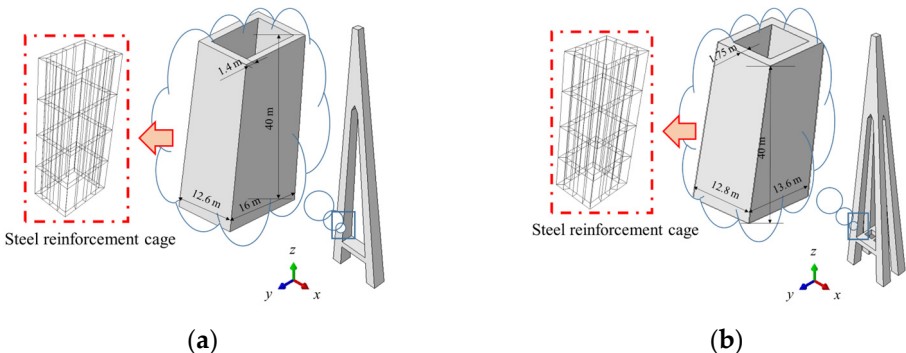

(**a**)  (**b**)

**Figure 1.** Structure of bridge pylon section: (**a**) SP and (**b**) MP.

**Table 1.** Fire scenario assumptions.

| Scenario | | Types of Scenarios | Location of Fire Source |
|---|---|---|---|
| I | 1 | Internal fire of SP | |
| | 2 | Internal fire of MP | |
| II | 1 | External ring fire of SP | |
| | 2 | External ring fire of MP | |
| III | 1 | External fire on one side of SP | |
| | 2 | External fire on one side of MP | |

### 2.2. Fire Scenario Simulation

2.2.1. Modeling Process and Parameters

For the impact of internal fire and external fire on the pylon column structure, 40 m high columns under construction of SP and MP are modeled to analyze the fire characteristics, as shown in Figure 2. Through sensitivity analysis, it has been determined that the temperature result error value remains within 3% when employing minimum grid values of 1.2 m, 1 m, and 0.8 m. In light of achieving optimal computational efficiency, the preferred minimum grid size is established at 1 m. Notably, the operational guidelines outlined in the FDS operation manual stipulate that the grid size should be 0.1 of the characteristic size of the fire source. Additionally, the relationship value of 0.083 provided in this paper further confirms the reasonableness of the selected grid value.

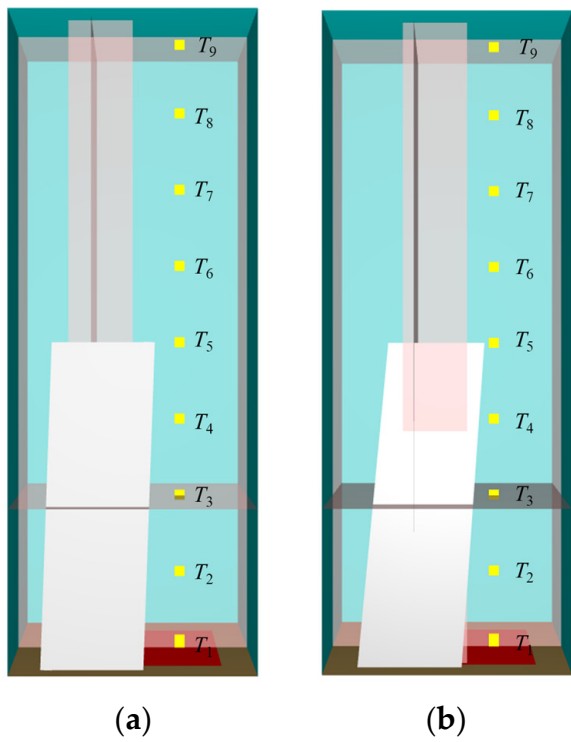

**Figure 2.** Column models under construction: (**a**) SP and (**b**) MP.

The sizes of the calculated spaces are determined based on the specific characteristics and geometric shape of the bridge pylon columns and fire source area. It should be large enough to cover relevant physical phenomena and boundary conditions while also small enough to maintain computational costs within an acceptable range. Therefore, the space sizes for numerical analysis of Scenarios I, II, and III are $20 \times 20 \times 80$ m$^3$, $25 \times 20 \times 80$ m$^3$, and $30 \times 20 \times 80$ m$^3$, respectively. The thermocouples named $T_1$–$T_9$ for monitoring temperature are set at heights of 1 m, 10 m, 20 m, 30 m, 40 m, 50 m, 60 m, 70 m, and 80 m, respectively.

2.2.2. Setting of Fire Sources

In the process of bridge construction, there are various kinds of combustible materials. In order to eliminate the differences in simulation results caused by the scale of the fire, the fire area in all scenarios is ensured to be consistent, all of which are 130 m$^2$. Although the types of combustible materials are different, the heat release rate (HRR) develops approximately according to the square law of time ($t^2$) at the initial stage of fire growth. The fire growth curve can be expressed as Equation (1) [31]. The combustion material is

thermal insulation foam. Foam combustion is a fast fire with a value of $\alpha$ 0.0469. The rising stage time of HRR can be calculated, after which the HRR remains stable.

$$Q = \alpha(t - t_0)^2 \tag{1}$$

where $Q$ is the heat release rate, $\alpha$ is the fire growth coefficient, $t$ is the time after the fire, and $t_0$ is the initial time.

## 3. Analysis of Fire Simulation Result

### 3.1. HRR

The total combustion area under the three scenarios is the same about 130 m², which is the bottom area inside the pylon column. The unit heat release rate of the foam is 0.273 MW/m² [32]. The time history curves of HRR for the different scenarios are given in Figure 3. The trend of the three curves is basically coincident. Therefore, it is proved that the fire source setting is the same and effective. It can be seen from Figure 3 that the HRR increases rapidly in the first 76 s, and then the HRR remains stable. This phenomenon is consistent with the development model of $t^2$ fire. The total simulation time is 1800 s.

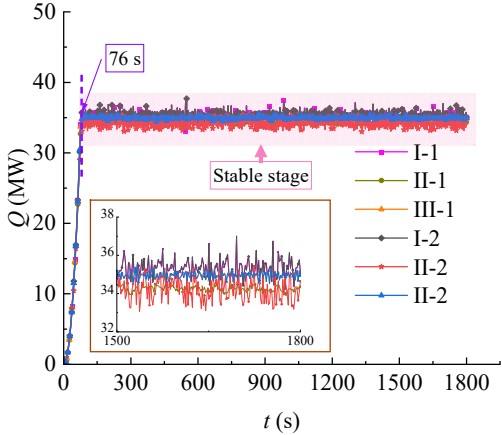

**Figure 3.** HRR results of different scenarios.

### 3.2. Temperature and Flue Gas Distribution

The SP temperature distribution contour plot at a randomly selected moment 205 s of the stable stage is plotted in Figure 4. Figure 4a shows the fire inside the pylon column, with smoke flowing from the bottom to the top. The fire position is located at the bottom of the pylon with high temperature. The fire has a greater impact on the inside of the pylon column and a smaller impact on the outside. Figure 4b is an external pylon fire that is relatively open. The flames wrap the outer ring of the pylon pillar in a ring, and all sides of the pylon pillar are overheated. The pylon column is wrapped in a ring of flame, and all sides of the outer surface are overheated. The smoke flows from the bottom to around. Similar to Scenario I-1, the temperature at the bottom of the pylon as the burning position is higher. In contrast to Scenario I-1, fire has a greater impact on the outside of the column but a smaller impact on the inside. Figure 4c is also an open external fire, which occurred on one side of the column. Smoke diffuses from the bottom of the pylon toward the surrounding column. Similarly, one side at the bottom of the pylon is where the fire burns and the temperature is higher. Different from other scenarios, the fired surface of the pylon column is greatly affected, and the unfired surfaces are slightly affected.

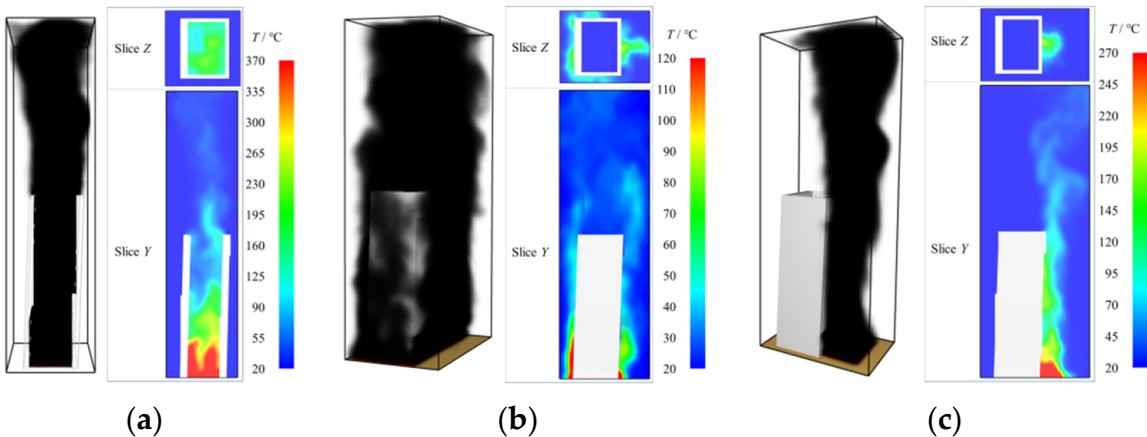

**Figure 4.** Temperature and smoke distribution of SP at 205 s: (**a**) I-1, (**b**) II-1, and (**c**) III-1.

Contrasting the maximum temperatures among the three scenarios, it can be known that the temperature in Scenario I-1 is the highest, that in Scenario II-1 is the lowest, and the temperature in Scenario III-1 is between Scenarios I-1 and II-1. The reason is that the fire in Scenario I-1 is in a relatively enclosed space, and the smoke can only escape from the top vent of the pylon column under construction. According to the theory of smoke stratification [33], heat accumulates in a large number of columns, resulting in higher temperatures. However, Scenarios II-1 and III-1 are open-space fires with stronger smoke mobility and less heat accumulation. In contrast, the fire area on one side of the pylon column in Scenario II-1 is smaller than that of Scenario III-1. Furthermore, the flame height and smoke volume in Scenario II-1 are also lower. As a result, the temperature of Scenario II-1 is the lowest. The simulation MP results of Scenarios I-2, II-2, and III-3 are similar to those of SP.

*3.3. Chimney Effect*

For hollow bridge pylons, the internal fire mode is significantly different from the external fire mode. Especially for the internal fire of a bridge pylon, the opening vent at the top and the working holes left at the bottom are connected to the internal channel of the bridge pylon under construction, forming a special structure similar to a chimney. The size of the working hole generally needs to accommodate at least one person for normal entry and exit, which is set to 1.25 m × 1.75 m. The main function of the working hole is ventilation. Compared to the overall size of the bridge pylon structure, the size of the working hole is small, which has limited mechanical impacts on the structure. Under the influence of the pressure differential between the interior and exterior of the bridge pylon, a significant volume of air is drawn in from the bottom. The combustion of combustibles is accelerated, and the smoke spreads rapidly upward along the vertical bridge pylon, resulting in the chimney effect.

In the simulation model of an internal fire, a worker manhole with a height of 2 m is added to the inner wall of the pylon. The hot smoke is floated via buoyancy from bottom to top, as shown in Figure 5. Due to the effect of internal and external pressure difference, the air outside the pylon is pushed into the fire space through the bottom lateral channel at a certain velocity. The combustion of the fire source is enhanced by supplemental oxygen. The fire plume is driven by buoyancy and starts to rise. With the continuous supplement of fire source energy and the continuous entrainment of external air into the pylon column, the volume of flue gas and air is increased after full mixing. Meanwhile, the gas mixture is gradually lifted under the driving buoyancy. During the rising process, the air is continuously mixed and involved, gradually filling the inside of the pylon column. After the column is filled with smoke, the gas in the column gradually reaches an equilibrium state. The external gas is involved in the bridge pylon from the bottom holes, and the internal gas is discharged from the top opening vent. Under the

continuous buoyancy drive and chimney effect [34,35], the flue gas in the bridge pylon is maintained stable without changing with time during the flow, which is a stable stage.

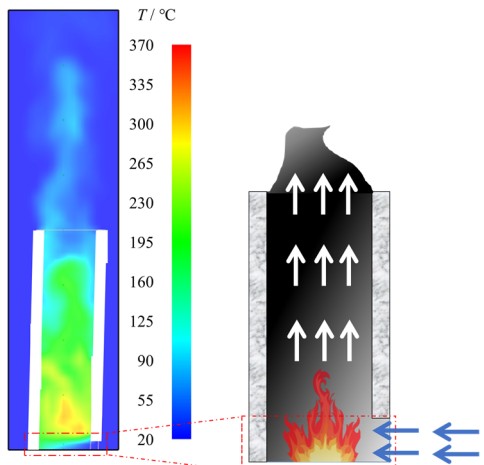

**Figure 5.** Chimney effect of pylon column.

### 3.4. Temperature Distribution

The measured temperature curves of nine thermocouples in the vertical direction are plotted in Figure 6. It can be seen that the trend of temperature rise is consistent with the $t^2$ model of fire. The values of temperature curves are not stable but fluctuate. The temperature fluctuation in Scenario I-1 is significantly higher than in Scenarios II-1 and III-1. The reason for this phenomenon is that the flame pulsation is more intense due to the enclosed space inside the pylon column. Therefore, the phenomenon of thermal convection is more intense, resulting in temperature fluctuations.

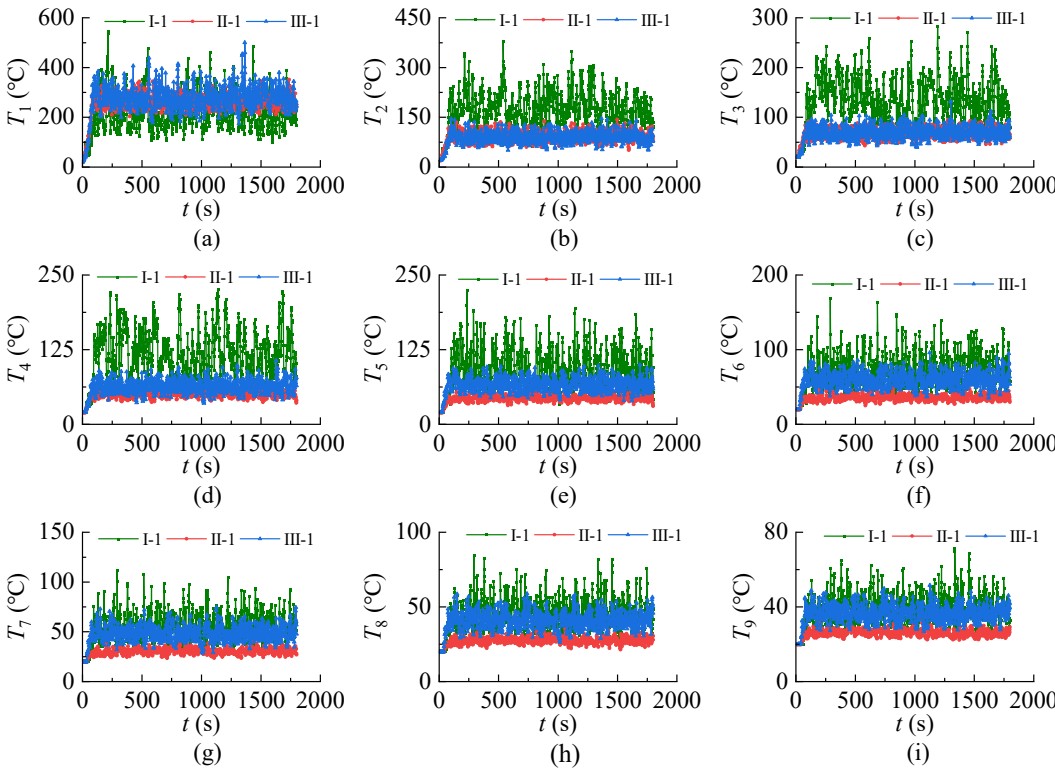

**Figure 6.** Thermocouple temperature results: (**a**) $T_1$, (**b**) $T_2$, (**c**) $T_3$, (**d**) $T_4$, (**e**) $T_5$, (**f**) $T_6$, (**g**) $T_7$, (**h**) $T_8$, and (**i**) $T_9$.

Figure 6a shows the temperature values near the fire source. Since the fire source has the same size and material, the average temperature values of the three scenarios are basically close to each other. Figure 6b,c are temperature values at 10 m and 20 m. Scenarios I have significantly higher temperatures than Scenarios II and III, while Scenarios II and III have almost the same temperature values. Because of the intense entrainment of fire in the pylon, the fire plume lifts the flame and extends the influence range of high temperature. Temperature values at 10 m and 20 m are plotted in Figure 6b,c. The temperature values of Scenarios I-1 are obviously higher than those of Scenarios II-1 and III-1, while the temperature values of Scenarios II-1 and III-1 are almost identical. Because of the intense fire entrainment in the pylon column, the flame is lifted by the fire plume, which extends the influence range of high temperature. Figure 6d–f describe the temperatures at 30, 40, and 50 m heights, respectively. It can be found that the temperature curves of Scenario III-1 are gradually higher than that of Scenario II-1 and approach that of Scenario I-1. The reason is that the fire source area on one side of the pylon in Scenario II-1 is smaller than that in Scenario III-1. Therefore, the flame height is lower, and the range of temperature effects in Scenario II-1 is more limited. Figure 6g–i are temperatures at 30, 40 and 50 m heights, respectively. It can be seen from the graph that the temperature curves of Scenarios I-1 and III-1 basically coincide and are higher than those of Scenario II-1. It can be explained that ring fires of the same size were affected by intermediate buildings, resulting in lower temperatures above the pylon pillars in Scenario II-1. Figure 6g–i give the temperatures at 30, 40, and 50 m heights, respectively. Conclusively, ring fires of the same size are affected by intermediate structures, resulting in lower temperatures above the column in Scenario II-1.

Under the influence of the $t^2$ fire, the temperature shows a trend of rapid rise and then stable fluctuation. The rising section of the temperature curve is the stage of fire development. The stable section of the temperature curve is the full combustion stage of the fire, at which the maximum temperature is basically reached. In order to observe the decay characteristics of the temperature with height increase, the mean temperature of 500–1000 s is taken for analysis, as shown in Figure 7. The temperature distribution in all scenarios decreases with increasing height. The trend of decreasing temperature rate from fast to slow is presented. At the same height, the temperature value of Scenario I-1 is higher than that of Scenario II-1, while the temperature of Scenario III-1 is between Scenarios I-1 and II-1. The temperature of Scenario III-1 in the range of 0–20 m is consistent with that in Scenario II-1. The temperature of Scenario III-1 is stable at the height of 20–60 m, which is basically unchanged. The temperature of Scenario III-1, with a height of 60 m, is gradually close to that of Scenario I-1.

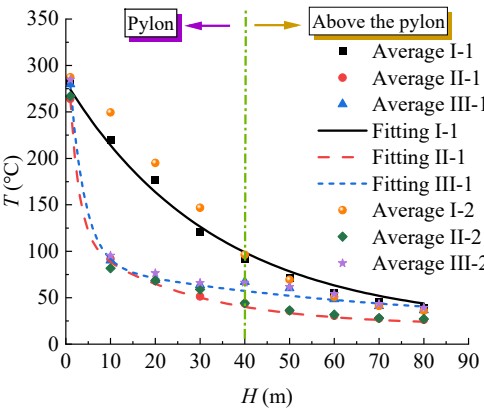

**Figure 7.** Variation of temperature with height.

The temperature distribution with increasing heights is fitted via a negative exponential function. The temperature point of Scenario I-1 is better fitted via a single negative exponent with a fitting rate of 0.9909. The $T_f$ and $\alpha$ in Scenario I-1 are 282 and 0.03, respec-

tively. Scenarios II-1 and III-1 are better fitted using two negative exponents with fitting rates of 0.9989 and 0.9958, respectively. The $T_f$, $\alpha$, $\mu$, and $\beta$ of Scenario II-1 are separately 264, 0.5, 100, and 0.04. The $T_f$, $\alpha$, $\mu$, and $\beta$ of Scenarios III-1 are 280, 0.3, 68, and 0.015, respectively. Vertical temperature field values for specific fire scenarios can be efficiently predicted using fitting functions.

$$T_h = \begin{cases} (T_f - T_0)e^{-\alpha H} + T_0 \\ (T_f - T_0)e^{-\alpha H} + \mu e^{-\beta H} + T_0 \end{cases} \tag{2}$$

where $T_h$ is the temperature of a certain height, $T_f$ is the initial smoke temperature, $T_0$ is the ambient temperature of usually 20 °C, and $H$ is the height value. In addition, $\alpha$, $\beta$, and $\mu$ are the fitting coefficients.

Under the same height condition, ignoring the difference in inclination angle, the SP and MP are only different in size. Scenarios I-2, II-2, and III-2 of the MP column are established to verify the effectiveness of Equation (2). Then, the calculated temperature values are basically similar to those in Scenario I-1 of SP, as shown in Figure 7. The results are consistent with the trend of negative exponential decline and highly resemble the fitted curve. Consequently, the fire temperature distribution of the MP and SP can be considered consistent, thus simplifying the temperature load boundary.

## 4. Thermal–Mechanical Coupling Analysis

### 4.1. Material Thermal Parameters

The concrete structure strength of the bridge pylon is C60, with a density of 2200 kg/m$^3$. The thermal conductivity and specific heat of concrete with increasing temperature are shown in Equations (3) and (4). At ambient temperatures, the elastic modulus of C60 is 36 GPa, and Poisson's ratio is 0.2. C60 is a high-strength concrete, and the high-temperature deterioration performance is more serious than that of ordinary concrete. The relationship between its elastic modulus and temperature is presented in Equation (5). The elasticity and yield modulus of steel bars are 210 GPa and 400 MPa, respectively.

$$\lambda_c = 1.68 - 0.19\frac{T}{100} + 0.82 \times 10^{-2}\left(\frac{T}{100}\right)^2 \quad (20\,°C \le T \le 1000\,°C) \tag{3}$$

$$\begin{cases} 900 & (20\,°C \le T \le 100\,°C) \\ 900 + (T - 100) & (100\,°C < T \le 200\,°C) \\ 1000 + \frac{T-200}{2} & (200\,°C < T \le 400\,°C) \\ 1100 & (400\,°C < T \le 1000\,°C) \end{cases} \tag{4}$$

$$\frac{E_{cT}}{E_c} = \begin{cases} 1.0 & (20\,°C \le T \le 80\,°C) \\ 2.24 \times 10^{-6}T^2 - 3.32 \times 10^{-3}T + 1.25 & (80\,°C < T \le 1000\,°C) \end{cases} \tag{5}$$

where $\lambda_c$ is the thermal conductivity coefficient of concrete with temperature variation, $c_c$ is the specific heat capacity of concrete with temperature variation, $E_c$ is the elastic modulus at ambient temperature, and $E_{cT}$ is the elastic modulus at high temperature.

### 4.2. Modeling of Finite Element

The pylon column model is established in the finite element (FE) software, with a size that is the same as that of the pylon column model in FDS. The FE models are meshed using structured grids. In order to improve the calculation speed of FE simulation, the models are reasonably divided. For the facing-fire surface, the meshed elements are denser in the thickness direction with a grid size of 0.025 m. At the same time, the elements of the back-fire part are rougher, and the size of the grids is 0.4 m. The number of grids per scenario is listed in Table 2. The dense positions of the model are one circle inside Scenario I, one circle outside Scenario II, and one side outside Scenario III. The types of thermal and mechanical calculation elements are DC3D8 and C3D8R, respectively. The bottom of the bridge pylon

is set as a fixed constraint during mechanical calculation. The ambient temperature is 20 °C. The calculation time is 1800 s. The similarity of the temperature field in the MP and SP has been confirmed in the fire simulation in Section 3.4. Therefore, in order to better compare the thermodynamic behavior characteristics of different structures, the temperature loads of the MP and SP are set identically in the same scene. The temperature loads are loaded in layers, and its non-uniform distribution is characterized by the average temperature of each height calculated via FDS.

**Table 2.** Number of grids.

| Scenario | Number |
|----------|--------|
| I-1 | 248,460 |
| I-2 | 256,000 |
| II-1 | 246,000 |
| II-2 | 256,000 |
| III-1 | 89,800 |
| III-2 | 88,700 |

## 5. Analysis of Thermal–Mechanical Coupling Results

### 5.1. Heat Transfer Analysis

The temperature loading of a non-uniform fire is shown in Figure 8. The high-temperature zone at the bottom of the column is observed magnificently along the wall thickness. It is known that the temperature decreases gradually along the thickness direction. The heat transfer thickness values of columns at 0–10 m, 10–20 m, 20–30 m, and 30–40 m heights are calculated, respectively. As a result, the heat transfer thickness in Scenario I are approximately 200, 175, 150, and 125 mm in sequence, while the heat transfer thickness in Scenario II are approximately 175, 150, 150, and 125 mm in sequence, and the heat transfer thickness in Scenario III are 200, 175, 150, and 150 mm in sequence. It is indicated that the fire temperature has an impact within a certain range of pylon column thickness. The walls of the pylon are affected more deeply by higher fire temperatures. Cracks are easily generated on the facing fire with uneven heating and cooling of the inside and outside pylon, which will cause damage to the outer surface of the column.

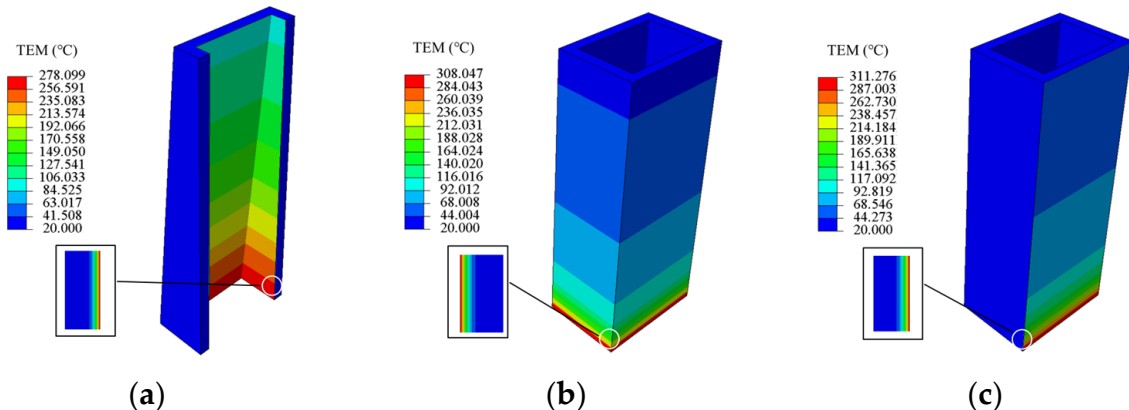

(a) (b) (c)

**Figure 8.** *Cont.*

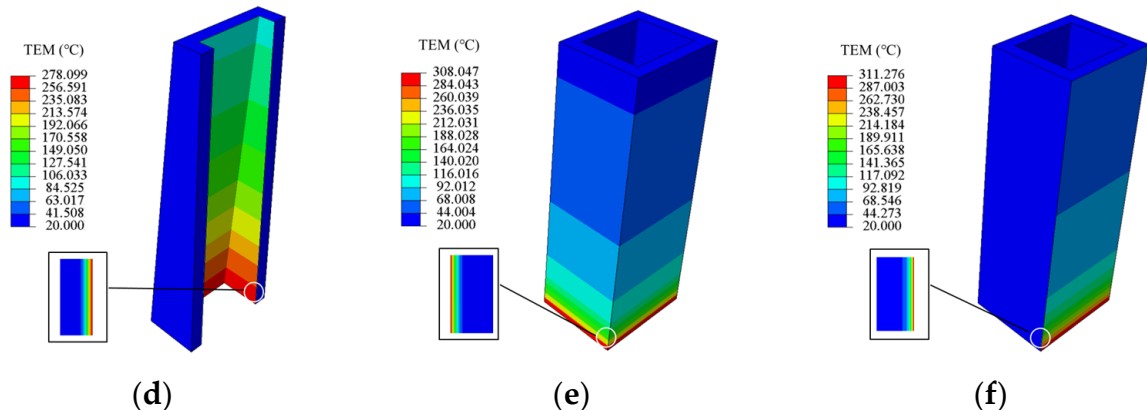

**Figure 8.** Heat transfer after temperature loading: (**a**) I-1, (**b**) II-1, (**c**) III-1, (**d**) I-2, (**e**) II-2, and (**f**) III-2.

With a dense grid thickness of 0.025 m as an interval, the temperature time history curves of different thicknesses are plotted in Figure 9. The outermost loading temperature is consistent with the trend of $t^2$ fire in FDS. At 25 mm thickness, the temperature at the end moment decreased by about 35%. The temperature curve is highly non-linear, and the rate of temperature rise changes from fast to slow. As the thickness increases, the temperature becomes closer to the ambient temperature of 20 °C. Moreover, the temperature rise curve gradually linearly increased. In addition, the starting point of temperature rise is also increasingly delayed as the thickness increases.

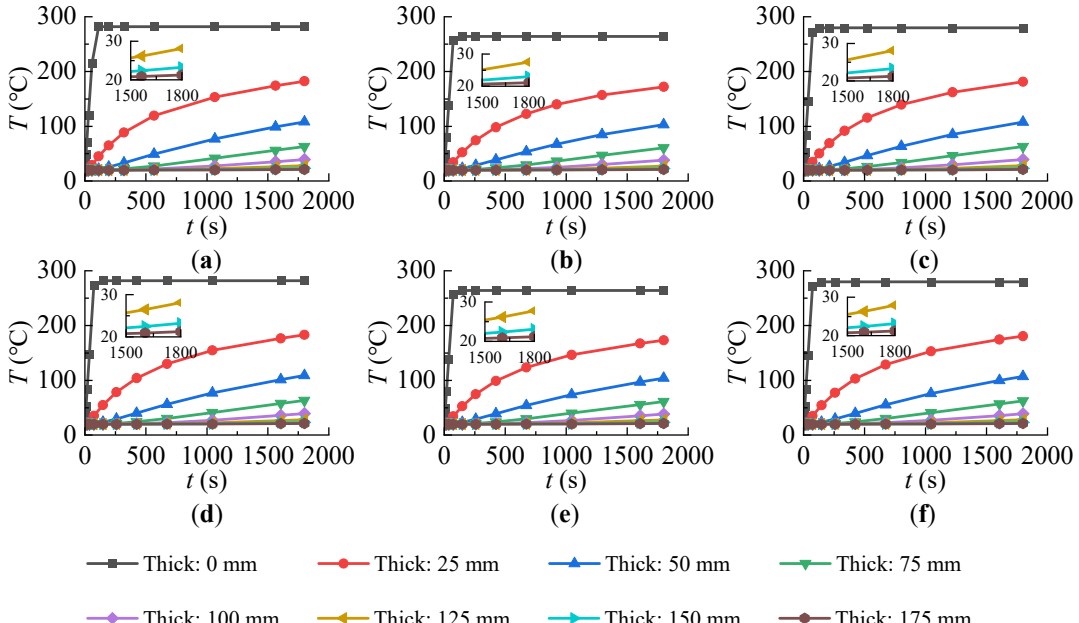

**Figure 9.** Temperature curves at different thickness: (**a**) I-1, (**b**) II-1, (**c**) III-1, (**d**) I-2, (**e**) II-2, and (**f**) III-2.

The time history curve of the highest temperature point on the steel reinforcement is plotted in Figure 10. It can be seen that the temperature of the steel bar under a 150 mm protective layer is very low, not exceeding 24 °C. In all fire scenarios, the steel bars are almost unaffected by the fire. However, if the steel bars are exposed due to concrete detachment, the impact of a fire on the steel bars will be significant.

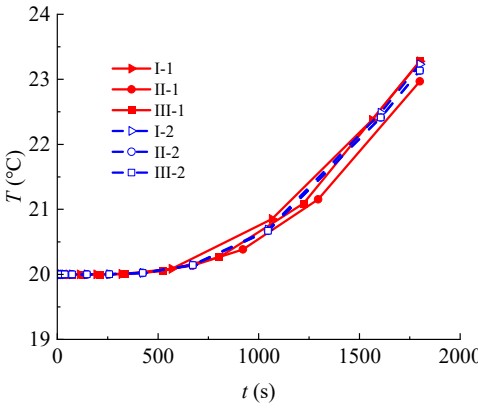

**Figure 10.** Steel reinforcement temperature.

*5.2. Mechanical Behavior Analysis*

5.2.1. Stress Field

The strength properties of concrete will decrease with increasing temperature. Under the action of fire, the tensile stress and compressive resistance of concrete should be reduced, as shown in Equations (6) and (7), resulting in structural damage to the pylon column. Due to the structural differences between the MP and SP, the inclination degree of the pylon column segments is also different. Assuming that the ground is in the *xoy* plane, the columns of MP are inclined towards both the *x*- and *y*-axes, while the columns of SP are inclined only towards the *x*-axis. The differences in the number and location of stress concentrations between the two pylon columns are caused by different inclination directions. Mises stress cloud plots for all scenarios are presented in Figure 11. There is one stress concentration point in the MP and two stress concentration points in the SP. The stress concentration positions are all located at the bottom sharp corners in the inclined direction of the pylon column.

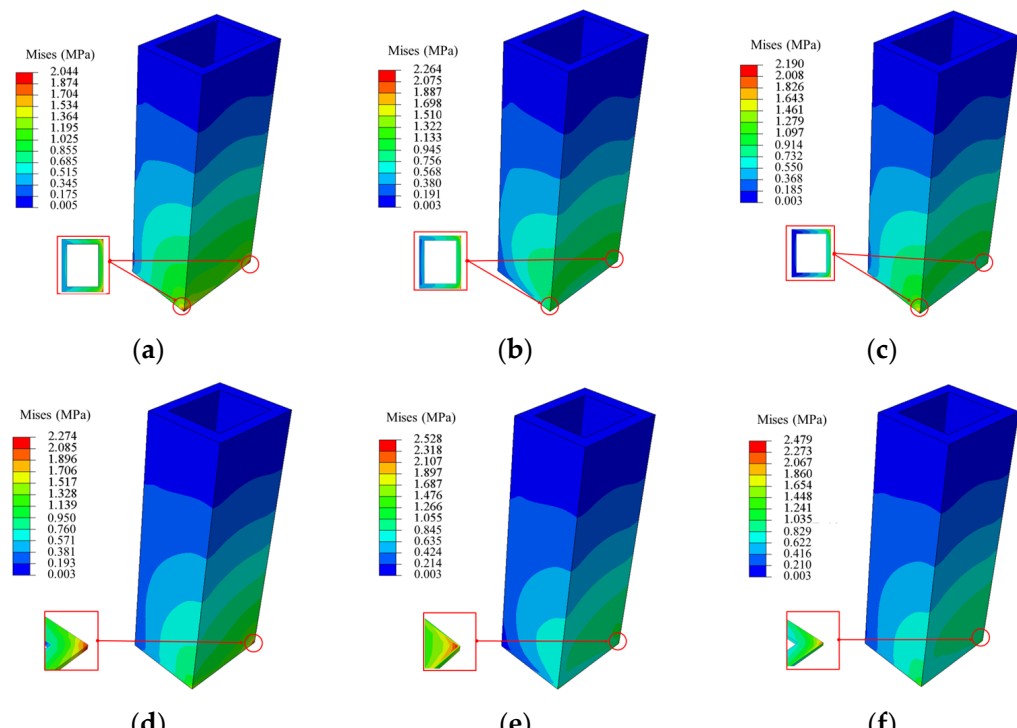

**Figure 11.** Stress distribution: (**a**) I-1, (**b**) II-1, (**c**) III-1, (**d**) I-2, (**e**) II-2, and (**f**) III-2.

In Scenario I, due to internal fire, the concrete at high temperatures undergoes deformation, which compresses outward and affects the external concrete. The stress distribution patterns are shown in Figure 11a,b. In Scenarios II and III, due to an external fire, deformations occur only within the range affected by temperature, leading to the stress distribution patterns shown in Figure 11b,c,e,f. In particular, the stress concentrations in Figure 11e,f are not generated at the outermost layer but at a thickness of 100 mm. In general, the stress distribution gradually decreases from bottom to top due to the influence of the temperature field. The stress value of the facing-fire side is greater than that of the back-fire side. Under the same scenario, the maximum stress of the MP is generally greater than that of the SP.

$$f_{cT} = \frac{1}{1 + 9.45 \times 10^{-8}(T-20)^{2.66}} f_c \qquad (20\,^\circ\mathrm{C} \leq T \leq 1000\,^\circ\mathrm{C}) \qquad (6)$$

$$f_{tT} = (1 - 0.001T)f_t \qquad (20\,^\circ\mathrm{C} \leq T \leq 1000\,^\circ\mathrm{C}) \qquad (7)$$

where $f_{cT}$ is the value of compressive strength varying with temperature, $f_c$ is the compressive strength, $f_{tT}$ is the value of tensile strength varying with temperature, and $f_t$ is the tensile strength.

The above analysis did not consider the thermal expansion coefficient parameter $\alpha l$. When considering the $\alpha l$ value of concrete as $1.05 \times 10^{-5}\,^\circ\mathrm{C}^{-1}$ [36], the stress value of the pylon columns increases exponentially, as shown in Figure 12. The stress value of concrete has changed from linear to nonlinear, and the maximum value has increased by about 27.5 times. The stress value rapidly increased in the first 80 s and then stabilized. In particular, the stress value on the overheated surface is the highest, far exceeding the standard compressive strength value of 27.5 MPa. Therefore, the calculation of the expansion coefficient is more conducive to fire protection design.

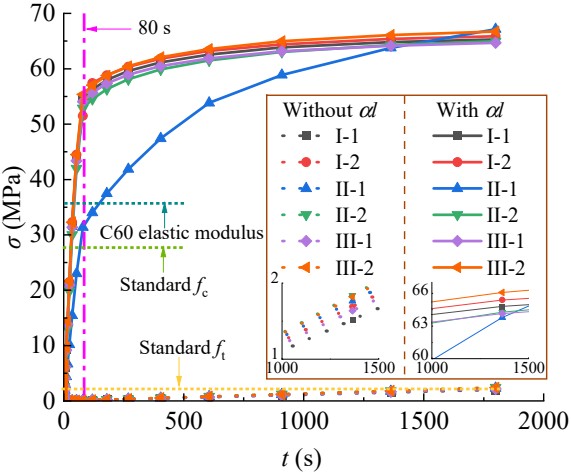

**Figure 12.** Time history curve of stress.

From all scenarios, it can be seen that the entire column is deformed by bending. The compressive stress is mainly borne on the inner side of the inclined column, and the tensile stress is mainly borne on the outer side of the inclined column. Concrete is a brittle material with a tensile strength far lower than compressive strength. Thus, the inner side is prone to compression failure at the stress concentration position of the column, while the outer side is prone to tensile failure.

### 5.2.2. Displacement Field

Because of the fixed restraint at the bottom of the column, no deformation occurs, while the free end at the top deforms with the development of fire. The calculation results of all the fire scenarios are enlarged by 5000 times and shown in Figure 13. The deformation of

the columns in all scenarios occurs in the inclined direction. This is also one of the reasons for the stress concentration at the bottom of the column. The maximum displacement of the SP is generated at the inclined side midpoint at the top of the column. The maximum displacement of the MP is generated on the inclined side edge at the top of the columns. The maximum deformation of the columns in Scenarios I-1, II-1, and III-1 is about 1.2 mm. However, that of Scenarios I-2, II-2, and III-2 is about 1.1 mm. The deformation of the MP is slightly higher than that of the side column, which indicates that stronger restraints are produced in the structure of the MP. This constraint is mainly from a thicker pylon wall, which means that the column is more stable with the greater wall thickness. These results without considering the thermal coefficient expansion are conservative.

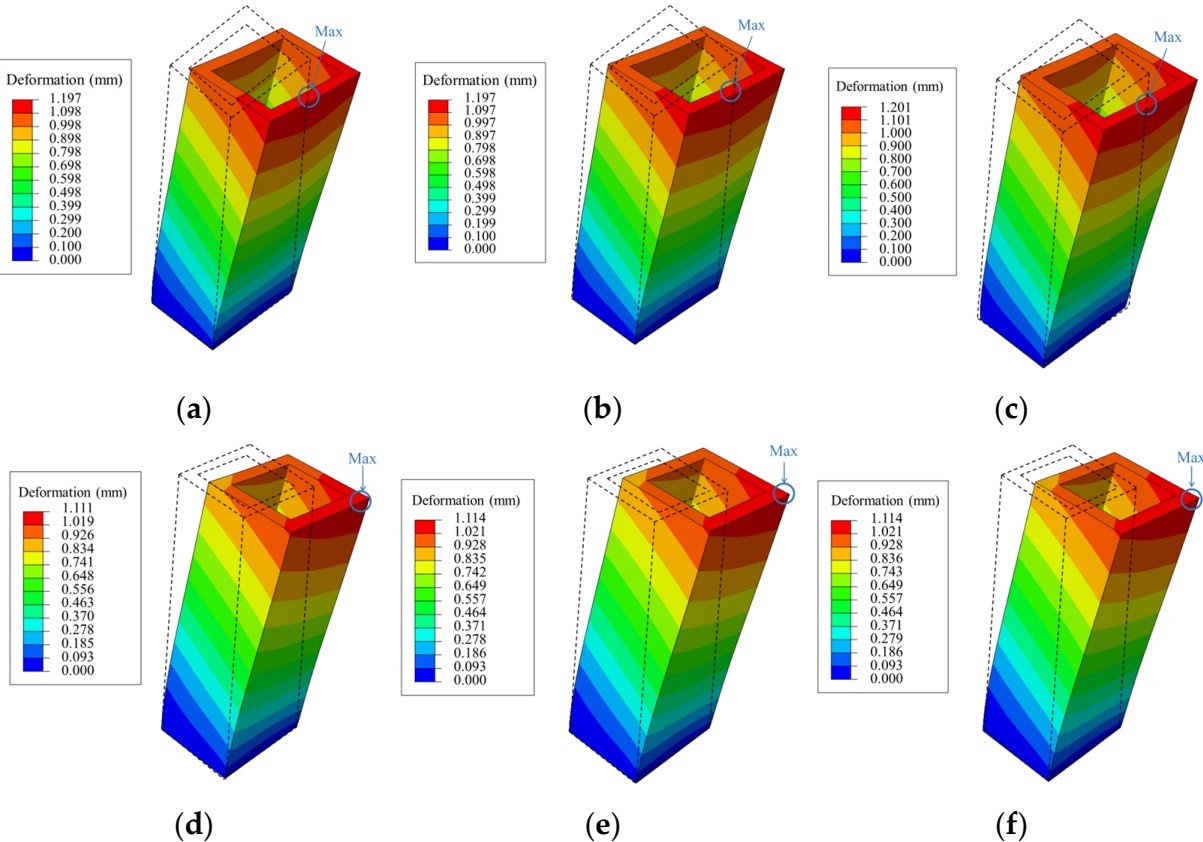

**Figure 13.** Deformation distribution: (**a**) I-1, (**b**) II-1, (**c**) III-1, (**d**) I-2, (**e**) II-2, and (**f**) III-2.

Figure 14 shows the displacement values of the pylon columns considering the thermal expansion coefficient. It indicates that the sensitivity of deformation to thermal expansion coefficient parameters is weaker than that of stress changes, and the deformation is 1–8 times the original value. The deformation trend is basically consistent with the change in stress value, showing a non-linear form of first fast then slow. Scenario II-1 of the SP and Scenario III-2 of the MP have the highest deformation. It is worth noting that Scenario III-2 exhibits deformation and retraction. The steel reinforcement and concrete deform harmoniously together.

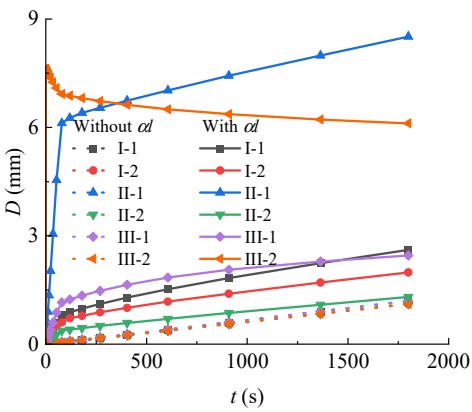

**Figure 14.** Time history curve of deformation.

## 6. Conclusions

The fire effects on a 40 m high bridge pylon under construction are explored in this paper. The mechanical behavior of the bridge pylon is analyzed in different fire scenarios using a thermo-mechanical coupling simulation method. The following conclusions can be drawn:

1. In the same bridge pylon, the temperature is higher in internal fire scenarios. Additionally, internal fire conditions promote the generation of the chimney effect, and the temperature influence extends in the vertical direction.
2. Along the vertical direction, a single negative exponential decline of temperature is observed in internal fires, while external fires exhibit a binomial negative exponential decline of temperature.
3. The transfer of fire temperature occurs in the direction of column thickness. The column is subjected to destruction within a specific thickness range. The maximum thickness affected in this study is 200 mm, which renders the column susceptible to cracking.
4. Under fire conditions, the column undergoes bending deformation in the inclined direction. The maximum deformation occurs at the top of the inclined inner side of the column. The bottom of the column experiences stress concentration points, with two stress concentration points in the SP and 1 in the MP, which are vulnerable to compression damage.
5. By comparing the simulation results of models with and without considering the thermal expansion coefficient, it was observed that the thermal expansion coefficient significantly impacts the model results, leading to higher stress and deformation values. Therefore, incorporating the simulation of the thermal expansion coefficient is crucial for enhancing the safety of fire protection design.

## 7. Prospects

In future investigations, a more comprehensive exploration will be pursued encompassing the following aspects:

1. A scaled test model will be established to delve into the fire phenomena occurring within the restricted spatial confines of the pylon columns, allowing for an in-depth analysis of the governing principles.
2. By conducting simulations and manipulating the length of the pylon column, the crucial length threshold for extinguishing internal flames will be determined, thereby aiming to identify the critical value.
3. Various configurations encompassing distinct positions, dimensions, and quantities of working holes will be established to investigate the fire impacts on bridge pylons featuring such apertures, thereby allowing for a comprehensive evaluation of their effects and implications.

4.  Fire sources with different materials and unit heat release rates are considered in bridge pylon fire simulation to obtain more conservative fire safety results.

These proposed avenues of research will contribute to enhancing the understanding of fire dynamics within bridge pylons and aid in formulating appropriate strategies for mitigating potential hazards.

**Author Contributions:** Conceptualization, Y.L. and Z.W.; methodology, Z.W. and C.W.; validation, Y.L. and Y.Z.; formal analysis, Z.W. and L.L.; investigation, H.M. and Z.W.; resources, Z.W.; data curation, H.M.; writing—original draft preparation, Y.L. and Z.W.; writing—review and editing, Y.L., Y.Z. and L.L. All authors have read and agreed to the published version of the manuscript.

**Funding:** This research was funded by the National Natural Science Foundation of China under grant No. 52278301.

**Institutional Review Board Statement:** Not applicable.

**Informed Consent Statement:** Not applicable.

**Data Availability Statement:** The raw data supporting the conclusions of this article will be made available by the authors, without undue reservation.

**Acknowledgments:** Financial support for this study was provided in part by the National Natural Science Foundation of China under grant No. 52278301. The results and opinions expressed in this paper are those of the authors only, and they do not necessarily represent those of the sponsors.

**Conflicts of Interest:** The authors declare no conflict of interest.

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
