# Peer review of "Fire Effect and Performance of Bridge Pylon Columns under Construction"

_fire, doi:10.3390/fire6100387_

Round 1
Reviewer 1 Report
1.Page 2, Line 69~72: The sentence here is repetitive.
2. Page 3,Line 113: Error! Reference source not found. There are multiple prompts for such errors in the text. Please check.
3.Page 3,Line 114: The introduction of thickness is inconsistent with that shown in Fig. 1. The size information displayed in Fig. 1 is incomplete. It is recommended to provide a more detailed description on the geometric dimensions, materials, and reinforcement information of SP and MP.
4.Page 3,Table 1:It is recommended to illustrate the location of the fire source in Scenario 2 and Scenario 3.
5.Page 4, Line 129: How is the space sizes determined?
6.Page 4, Line 138: Is the inner bottom area of the side pylon the same as that of the main pylon?
7.Page 5, Fig.3: According to the formula in Equ. , the heat release rate keeps increasing. Why does the subsequent temperature show a stable stage?
8.Page 6, Line 186: What is the size of the working holes? Has the influence of the opening been taken into account when modeling the finite element model?
9.Page7, Line 218:Scenarios have significantly higher temperatures than Scenarios 2 and 3?Line 237~239, The sentence here is repetitive.
10.Page8, Line 249: It is recommended to illustrate the specific location of the mid section with a diagram.
11. Page9, Line 269: What does III-3 refer to?
12.Page 9,line 279:Density and thermal expansion coefficient are 2200 kg/m3?
13.Page 9:Is the pylon reinforced with steel bars? Is the influence of reinforcement considered? How do the mechanical properties of materials vary with temperature? Equ. 6 and 7 on page 13 show that the compressive strength of concrete varies with temperature. It is recommended to add explanations in the main text.
14. Page 10: It is recommended to explain the meaning of heat transfer depth.
15.Page 13: There is an erroe in the captions of Fig10. The outer concrete at normal temperature is queued?
16.Page 13:LThere is an error in the symbol interpretation of L363; L368~372:What kind of fire scenario is this paragraph targeting?
17..Page 14: It is better to use dashed lines to display the position before deformation in the figure, so as to better visualize the deformation situation.
18.Page 14: What does the stress-strain relationship in section 5.2.3 refer to? The content of this section seems meaningless. What do stress and strain specifically refer to in a three-dimensional state? The slope of the curve in the figure is different, possibly because the temperature is different.
The english expression can be improved appropriately.
Author Response
Thank you very much for taking the time to review this manuscript. Please find the detailed responses below and the corresponding revisions in the re-submitted files.

Reviewer 2 Report
All the comments are as follows:
-The abstract of the manuscript does not reflect the main findings of the work. I suggest presenting some results quantitatively.
- Please revise all the "Error! Reference source not found" and put the correct Figures/tables accordingly.
- I can't find the citation for references [20 and 21]. Therefore, please rearrange all the References accordingly.
-The English write-up of the paper requires overall editing.
i.e: Scenario III is a fire a fire caused by accumulation on one side outside the 120 bridge pylon column.
-It would be good if the authors could add a future scope section with a proposed application after the conclusion section.
-The English write-up of the paper requires overall editing.
i.e: Scenario III is a fire a fire caused by accumulation on one side outside the 120 bridge pylon column. Please rephrase the sentence.
Author Response

(The authors gave the same response as above.)

Reviewer 3 Report
The paper deals with an interesting topic, however, the execution in not sufficient and it the manuscript should not be published until substantial revisions and clarifications have been made.
There are several unsupported claims and missing data sources, e.g.:
In recent years, the probability of fires occurring in bridge engineering has gradually increased.- no supporting evidence
Equations expressing relationships between temperature and material properties.
Material properties in section 4.1.
The fire model as a source of thermal effects is poorly described - grid resolution is not mentioned, no sensitivity analysis is performed, no explanation is given to the fire area, etc.
The temperatures profiles are measured at rather coarse intervals - 10m apart. Given the fluctuation of the temperature are thermocouples or gas temperature commands used for recording the actual temperatures? In addition, given the size of the fire, the predicted temperatures appear to be rather low - below 400 °C which is of concern.
Following on, it is unclear, how were the fire (temperature prediction) and mechanical models were coupled, i.e. how was the heat transfer from gas to solid modelled.
The overall deformation of 1.2mm and 1.1mm appears to be insignifficant given the dimensions of the structure modelled .
In additon, concrete retains most of its mechanical properties at indicated properties. Looking at the graphs in figure 9 no significant deterioration is expected.
Summarising the above, the fire model should be reviewed thouroughly, as the results presented indicate rather low temperatures, which may underestimate the potential effects on the exposed structure.
Although the paper flows relatively well, there are some expressions that are ambiguous or not standard, e.g.:
near-tower fire scenarios - ??
combustible species - combustible materials
rainbow diagram - slicefile / countour plot
air is inhaled from the bottom
air is continuously mixed and involved up
t2 law - t2 model
more unstable temperature
law of temperature attenuation - no such thing exist, this is caused by air entrainment into the smoke plue as it rises above the fire source
is observed magnificently along
etc.
Author Response

(The authors gave the same response as above.)

Round 2
Reviewer 1 Report
The paper has been revised or explained according to the review comments, so I suggest it can be accepted now.
Reviewer 2 Report
After downloading the revision of the full manuscript paper which has been submitted by the author for review. Some of the "Error! Reference source not found" can be noticed. Therefore, please double-check and revise all the "Error! Reference source not found" carefully and put the correct citations.
Based on the previous comments, this full manuscript paper can be accepted for publication after revising the comments accordingly.